# The Religiousness of Cultivation in the *Zhuangzi*: "The Unity of Self" of *Zuowang* 坐忘

**Shanshan Ma**

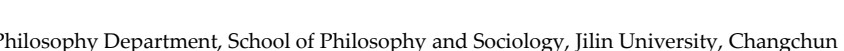

Philosophy Department, School of Philosophy and Sociology, Jilin University, Changchun 130012, China; mass19@mails.jlu.edu.cn

**Abstract:** From the perspective of mysticism, the interpretation of *zuowang* 坐忘 as the breath-meditation technique with a transcendental goal establishes the religious basis of Zhuangzi's cultivation. In contrast, most Chinese scholars argue instead that *zuowang* is primarily the mental state independent of meditative techniques, and that the techniques are devoid of philosophical significance. The pivot of the two seemingly irreconcilable views is the holistic concept of *qi* 氣. Combining the two interpretations can lead to a consummate understanding. There is an innate connection between the physical and nonphysical *qi*, and the latter can be considered as the root and basis. Deep exploration of the mysticism paradigm represented by Harold Roth reveals that the interpretation of meditative techniques is a necessary step in gaining mystical experiences not contradictory to the one that construes it as mental cultivation. The mind–body transformation shows that the pursuit of authenticity and wholeness of humanity can broaden one's concern with fellow human beings and other realms of existence, as the same process is followed in the perpetual growth and change of the universe. This allows us to experience resonance with the universe toward the goal of "the unity of self" so to speak. The religiousness of Zhuangzi's cultivation is, thus, substantiated.

**Keywords:** the *Zhuangzi*; *zuowang*; cultivation; religiousness; *qi*; the unity of self

## 1. Introduction

From the perspective of mysticism, rethinking the religiousness of the *Zhuangzi* 莊子 around the concept of *zuowang* 坐忘 is the key to understanding the nature of Zhuangzi's thought and its ultimate goal. A common core feature of the world's mystical traditions is the experience of "oneness" with the Ultimate Being through specific meditation practices. Many scholars have defined *zuowang* of the *Zhuangzi* as a contemplative technique of Daoist quiet sitting. They think that the idea of "merging with the Great Pervader" (*tong yu datong* 同於大通),[1] i.e., "the unity between humans and the Dao", is the mystical experience obtained through some meditation practices. Therefore, they concur it is an important criterion for judging Zhuangzi's theory as having a religious nature arising from mysticism.[2]

However, even if there are meditation-related contents in the *Zhuangzi*, can its religious attributes, thus, be confirmed? Moreover, many scholars do not agree that the interpretation of *zuowang* as a meditation technique captures its true meaning. In that case, does the point of the so-called religiousness of Zhuangzi become meaningless? A more meaningful question to explore, in my view, is the specificity of Zhuangzi's mystical experience in terms of meditation practices compared to Western religious mysticism. In the past century-old study of Zhuangzi's mystical philosophy, the unique character of Zhuangzi's cultivation in terms of *zuowang* remains a key question worth exploring.[3] In this study, I try to clarify whether the idea of *zuowang* has religiousness embedded in it by focusing on the key factor of Zhuangzi's thought that tends toward the mystical.

## 2. Meditation Practices in the Interpretive View of Religious Rituals

Around the topics of mysticism, many scholars in the European and North American sinological circle have been focused on the relationship between the *Zhuangzi* and the primitive religion (i.e., shamanism), paying attention to the religious dimension of Zhuangzi's thought. To be clear, the "mysticism" or "mystical experience" they focus on here mainly refers to the meditation rites highlighted in the *Zhuangzi*, which are mainly reflected in the several metaphors of *zuowang*, "fasting of the mind" (*xinzhai* 心齋), "guarding the One" (*shou yi* 守一), and "gathering *qi*" (*ji qi* 集氣).

Many sinologists, influenced by the Daoist studies of Henri Maspero (1883–1945), have shown that the meditative element forms an essential part of Zhuangzi's thought. In the article "The Saint and the Mystical Life in the *Laozi* and the *Zhuangzi*", Maspero concludes that the utmost prominent feature of Daoism (which he calls "Ancient Daoism") in Lao-Zhuang's era is "mystical life practices" (pratiques de vie mystique), which are, in fact, breath-meditation practices (Maspero 1950, p. 231). In particular, he takes *xinzhai* as a typical example of meditation in the *Zhuangzi* and summarizes the characteristics of this mystical practice into several primary stages:[4]

> "Thus, the mystical experience dominated their entire philosophy. From it, they derived what was original in their metaphysics; from it, they got their psychology, and finally, it furnished the principles for their philosophy of government." (Maspero 1978, p. 310)

Overall, the mystical aspect of Daoist meditation practices lies in the fact that it can bring people a unique spiritual experience in the sense of "unity with the Dao".

In the history of European sinology, Maspero first makes the point explicitly that the meditation rituals are the primary criteria for identifying early Daoism. He previously made a clear inference to the Lao-Zhuang School in the Warring States period as a group of masters and disciples who inherited mystical techniques. In his opinion, Laozi, Zhuangzi, Guan Yin 關尹, Liezi 列子, and Qu Yuan 屈原 (ca. 340–278 BCE) represented a mystical practice school focusing on spiritual cultivation among the ancient Daoist schools. His foresights deeply influenced later sinologists, especially Daoist researchers' thinking on the religious dimension of Zhuangzi's thought. Moreover, he traces the historical origin of this quasi-meditation practice which is unique to Lao-Zhuang Daoism. He assumes it to be a primitive religion that shares the same root as Daoism after the Six Dynasties period.[5] The importance of this viewpoint lies in the connection between the so-called "Daoist philosophy" and "Daoist religion" based on the meditation practices, which he links at the roots to the religious ritual of primitive religion or shamanism, particularly the practice of "fasting" (*zhai* 齋). This relates to the critical issue of the religiousness of Zhuangzi's thought. Compared with primitive religion and, later, religious Daoism, Zhuangzi's philosophy is related but has its unique characteristics. As for what these aspects entail, Maspero does not elaborate in his discussion of Lao-Zhuang's mystical thoughts. Nevertheless, his conclusion at least suggests that the interpretation of *zuowang* in the *Zhuangzi* as a specific practice of breathing meditation has its roots in history. It is not necessarily a far-fetched adaptation of the commentators, such as Cheng Xuanying 成玄英 (fl. 631–650) and others with a Daoist religious affiliation, who interpreted Zhuangzi's thought according to Zen or later Daoism.

Harold Roth's treatment of Daoist mysticism inherited Maspero's theory of mystical practice, and he created a set of distinctive doctrines that greatly influenced sinology in Europe and America. Roth's publication of *The Contemplative Foundations of Classical Daoism* (Roth 2021) brings together the most important ideas through his 30 years of research into the historical origins of Daoism. The conventional view of the so-called "Daoist school" is that its name and substance were artificially constructed by historians of the Two Han dynasty. However, Roth's archaeological identification of the Daoist literature argues that, although it did not have the name "Daoism" in the pre-Qin dynasty, the Daoist school existed as a group of teachers and disciples who practiced breath meditation:

> "The very fact that coherent contemplative practices were identifiable from text to text had to imply the existence of certain social organizations, lineages of teachers and students who taught and studied the practices and kept them alive across the generations until more organized and readily identifiable religious institutions formed at the end of the Han dynasty." (Roth 2021, p. 7)

Although Roth is aware that the lineage of the "Daoist school" is not as clear and strict as that of later religious Daoism, he insists that even such a loose spontaneous group must have some intergenerational inheritance relationship. Because of this, we can see many important ideas and concepts closely related to the "techniques or tradition of the Way" (*Daoshu* 道術) throughout the history of the Daoist literature.

Significantly, Roth traces the Daoist techniques to a set of quasi-meditation practices with elements of trance and fasting described in the most ancient text ever found, "Inward Training" (*Neiye* 內業) in the *Guanzi* 管子. It is considered that the *Zhuangzi* is the most detailed extension of this meditative technique, and that almost all-important concepts of Zhuangzi's thought can gain a more tangible and profound understanding from the embodied experience of meditation practices.[6] Roth elaborates a series of contemplative practices presented in the *Zhuangzi*, which can be more succinctly summarized as the following stages: "aligning the limbs in the proper sitting position", "cultivating and refining the breath", "getting rid of self-consciousness", "attaining stillness or tranquility", and finally, "developing the transformed cognition and achieving enlightenment".[7]

As for philosophical inquiry, Roth does not go much further than ancient sources of ideas related to meditation practices in the *Neiye* and the *Zhuangzi*. Nevertheless, he does point out the direction for us to rethink of the complicated connection between shamanism and the "axial breakthrough" of pre-Qin Daoist thoughts. On this issue, Yang Rur-bin 楊儒賓 goes further:

> "The shamanism hypothesis is extremely helpful to our understanding of the origin of Daoist thought, and its explanatory power is much greater than that of a generalized primitive religious ascription or the rationalized philosophical explanation." (Yang 2019, p. 135)

Focusing on the overall developmental lineage of Daoism, Yang argues that the "methods" (*shu* 術) of "emptiness" (*xu* 虛) and "tranquility" (*jing* 靜) in Daoist Classics (mainly the *Four Texts of the Yellow Emperor* 黃帝四經, the *Laozi*, and the *Zhuangzi*) is likely derived from the ancient experience of fasting ceremonies practiced by shaman officials. He points to this religious practice of fasting ritual as the essential element that united early Daoist movements into one stable tradition (Yang 2019, pp. 146–49). To a certain extent, his assertion supports Roth's definition of the nature of Zhuangzi's thought as "contemplative Daoism". It shows the significant and subtle connection between Zhuangzi's philosophy and primitive religion.

If we can accept Maspero and Roth's interpretation of *zuowang* as a meditation ritual, does it mean that the *Zhuangzi* is a quasi-religious text as many sinologists have suggested? I think the issue of Zhuangzi's religiousness is more complex than that. Undeniably, Zhuangzi's treatises on rituals and ceremonies contain some religious phenomena, as shown in worldwide presence of shamanism. However, Zhuangzi's thought, as an instance of the "philosophical breakthrough" of the Axial period, has undergone significant transformations compared to ancient shamanism. The recognition of *zuowang* as "trance" and "fasting" rituals shows the profound influence of shamanism on Zhuangzi's thought. However, we cannot simply use shamanism or religion to summarize early Daoism's mental state and its unique practices. In my view, the ultimate Dao in the *Zhuangzi*, as the flexible power that infuses the entire cosmos, is on par with the ancient shamanic deity; it can integrate and transform the primitive shamanic elements into itself, and it also has the ultimate concern that the Godhead achieves. Basically, Zhuangzi's religious thought is a kind of awakened humanism and anthropologism. It is the full manifestation of human initiative and potential in the light of the Dao, which directly suggests that Zhuangzi has

critically transformed and inherited the primitive way of meditation. If combined with the later institutional Daoist meditation, this point more clearly shows the uniqueness of the religious nature of Zhuangzi's conception of meditative cultivation.[8]

In philosophical studies, many also agree that metaphors such as *zuowang* and *xinzhai* are prototypes of the later Daoist practice of "internal alchemy" (*neidan* 內丹). In the narrow area of Daoist studies, the tendency to interpret *zuowang* as a meditation technique of breath cultivation for health and longevity is even more pronounced. Since the meditative practice in both later Daoism and the *Zhuangzi* could be considered a form of embodied and ritualistic practice, are there any subtle differences behind their formal similarities? In other words, can we define the religiousness of Zhuangzi's theory of cultivation simply on the basis of empirical criteria regarding the techniques of a meditation ritual? In my opinion, between later Daoism and the *Zhuangzi*, there must be a difference. For example, it is difficult to regard the theory of *zuowang* of Sima Chengzhen 司馬承禎 (d. 735) as a reproduction of the meaning of Zhuangzi. Merely identifying the similarities between the two in terms of meditation technique is far from revealing the uniqueness and significance of Zhuangzi's religiousness. Benjamin Schwartz (1916–1999), who also followed the mysticism paradigm of early Daoism, was keenly aware that the "mystical technique" of meditation is not the core of Lao-Zhuang, arguing that what matters is "the vision of reality" achieved at the cognitive level through such activity:

> "Yet in dealing with the major mystical orientations, such as Brahmanism, Sufism, or the mysticism of Meister Eckhardt, I find that it is the vision of reality that corresponds to mystical Taoism and not the description of the technique." (Schwartz 1985, p. 199)

Even though Roth defines the nature of early Daoist philosophy as a form of contemplative practice, his understanding of such meditation practices is not dogmatic. To be precise, Roth redefines the technique of Daoist meditation in terms of the Dao on the ideological level:

> "The distinctive ideas of inner cultivation begin and end with the Way or Dao 道 as the ultimate source of the cosmos and potency (*De* 德) as its manifestation in terms of concrete phenomena and experience; nonaction (*wuwei* 無為) as the definitive movement of the Way, and formlessness (*wuxing* 無形) as its spatial mode are both essential as well. There is also a common self-cultivation vocabulary that includes such results as stillness and silence (*jimo* 寂漠), tranquility (*jing* 靜), emptiness (*xu* 虛), and a variety of 'apophatic' or self-negating techniques and qualities of mind that lead to a direct apprehension of the Way." (Roth 2021, p. 6)

The religiousness of Zhuangzi's theory of cultivation engenders the profound humanistic care for human nature and the nature of all things, rather than simply a world system of institutional and ritualized religion. As the *Zhuangzi* shows, the forms of meditation techniques are not limited to the systematic or institutional norms of religion (like many sects of later Daoism) but reorganize themselves in a very broad and flexible way that breaks the traditional boundaries of religion, organization, and social class. Once we master the inclusiveness and flexibility of Zhuangzi-style meditation, we find that there are a large number of diverse underprivileged personas in society (especially craftsmen) that present contemplative concentration. They are described as having a deep state of stillness and silence, such as a withered tree or slaked lime, which can be regarded as achieving the ideal state of Zhuangzi-like meditation practices. All kinds of activities of daily living, such as Cook Ding carving an ox, the cicada catcher catching insects, the skilled swimmer going over the falls, or the bellstand carver chipping wood, could be considered as the way of "embodiment of the Dao" or the "technique of the Dao". They can adapt their personal living conditions, fulfilling the potential of their unique nature, where they manifest the ultimate pursuit of "unity with the Dao" in all aspects.

Regarding the humanistic care embodied in ritualistic techniques, the spirit of Zhuangzi's theory may be close to the French sociologist Émile Durkheim's definition of religion. Neither are confined to institutionally and doctrinally narrow views of religion. Generally speaking, Durkheim's definition of "religion" breaks the conventional view prevalent before the 19th century. In his redefinition, "sacredness" (or "sanctity") is considered the most fundamental attribute; rituals and clear rules of behavior associated with the sacred are regarded as the second element, whereas specific organizational groups are defined as the third element (Durkheim 2012, p. 47). The first and most core principle of "sanctity" mentioned here resonates with Zhuangzi's view of the transformative effect of meditative cultivation to a considerable extent. That is, its aim is to restore the original nature of humans and all living things, to fulfill the originality of everything itself, and to restore the true sovereignty of our lives through certain practical activities. It is not about yielding to the external determinacy of some given conditions, but allowing all living individuals to move toward a state of complete freedom. In this regard, Zhuangzi particularly criticizes the stranglehold on human nature by the "propriety" (*li* 禮) of social norms, which affects the creativity and imagination of countless lives. In my view, both Zhuangzi and Durkheim reinterpret the sacred value of religion with a tolerant attitude, particularly as both of them go further into the fundamental of human potentials. This is elucidated in more detail later in this article.

Overall, from the ritual perspective, Zhuangzi's meaning of *zuowang* may refer to a form of meditation practices. It should be noted, however, that an important religious element makes this cultivation markedly different from that of primitive religion (i.e., shamanism) and later Daoist rituals. In short, when it comes to the religiousness of Zhuangzi's thought, the key lies not in the experiential description of the meditative rituals or techniques, but in the humanistic concern of the mystical experience that Zhuangzi pursues through meditation practices. In other words, the specificity of Zhuangzi-like meditation is not about the technique itself, but the perspective from which it is viewed.[9] In this way, by shifting from the entanglement of the phenomena to the difference in perspectives, we may be able to distance ourselves from those historical relics that cannot be examined.

## 3. De-Meditation and the Problem of the Spiritual-Cultivation View

The term of *zuowang* initially occurs in a separate dialogue between Yan Hui 顏回 and his teacher Confucius in Chapter 6 of "The Great and Venerable Teacher" (*da zong shi* 大宗師). Reviewing the interpretations of *zuowang* by successive commentators throughout history, there are basically two opposing views. One points to a practical technique of "sitting and forgetting", while the other points to the mental or conscious state of "unintentional forgetting". Commentators who held the meditation technique view include Cui Zhuan 崔譔 (ca. 265–317), Cheng Xuanying, and Guo Qingfan 郭慶藩 (1844–1896). In contrast, Yan Fu 嚴複 (1854–1921), Ma Xulun 馬敘倫 (1885–1970), Qian Mu 錢穆 (1895–1990), Wang Shumin 王叔岷 (1914–2008), and Zeng Guofan 曾國藩 (1811–1872), etc. advocated the view of metaphorical "de-meditation". The former view is also prevalent in European and American Sinology (particularly mysticism interpretations), while the latter reflects the dominant view of most Chinese scholars. It seems difficult to reconcile these two interpretative paradigms. In fact, both native and overseas scholars are more supportive of the latter view, contending that the meditation interpretation fails to capture the metaphorical characteristic of Zhuangzi's philosophy. In short, this simply materializes Zhuangzi's figurative statements.

Wu Genyou ( 吳根友) is the foremost representative I can see among Chinese scholars who has made an informative and influential interpretation of *zuowang* in the *Zhuangzi.* His claim of "forgetting without intentional actions" typically represents the dominant view (Wu and Huang 2017). To summarize his rebuttal against the interpretation of "sitting and forgetting", as meditative praxis, three main points can be proffered. First, on the semantic level, it denies that the term *zuo* 坐 of *zuowang* has the meaning of the verb "to sit up". It contends that *zuo* is an adjective of "natural" (or an adverb of "naturally") which has no



practical connotation, and this interpretation is more consistent with the meaning of this phrase in the whole text of the *Zhuangzi* and other contemporary texts.[10] Secondly, in terms of the commentary tradition, he enumerates many commentators since the Wei-Jin period, such as Kong Yingda 孔穎達 (574–648), Du Guangting 杜光庭 (850–933), Sima Chengzhen, Zeng Zao 曾慥 (?–1155), Lin Xiyi 林希逸 (1193–1271), and Wang Fuzhi 王夫之 (1619–1692), who misinterpreted the meaning of *zuowang* as the technique of meditation, as a result of the influence of the external culture of Zen sitting practice. Thirdly, from the central idea of the whole, he points out that the intentional practice of breath meditation is incompatible with Zhuangzi's philosophical goal of "nonaction" (*wuwei* 無為). These three arguments of Wu arise from different points of view, but they all try to argue that the idea of *wang* 忘 of *zuowang* means a natural state of mind, which has nothing to do with the meditative technique of quiet sitting or sitting meditation.

However, further reflections on these three points reveal some difficulties with the "de-meditation" interpretations. First of all, by looking deeper into the semantics of *zuo*, it can be found that the adverb "naturally" (or the adjective "natural") is still based on the practical implication of the verb root "sit". Wu cites the *Book of Rites* (*Liji* 禮记) as saying "大夫不坐羊, 士不坐犬", and he maintains that the meaning of *zuo* here should follow the interpretation of "without actual acts" (i.e., naturally) as annotated by Zheng Xuan 鄭玄 (127–200). Contrary to his opinion, I argue that the exact explanation should be just as said by Ling Shu 凌曙 (1775–1829): "no sheep shall be killed and sit on its skin if there is no justice cause, no dog shall be killed and sit on its skin for the cause 無故不得殺羊坐其皮, 無故不得殺犬坐其皮" (Zheng et al. 1999, p. 1417). That is to say, the term *zuo* in the sense of *zuoyang* 坐羊 and *zuoquan* 坐犬 should be interpreted as "sit" as an actual verb, not what Wu calls an adjective term without practical meaning. The highly flexible semantics of Chinese characters manifests the interplay between the empirical meaning of "sit" and its metaphorical, symbolic meaning, where the figurative and nonfigurative terms are both opposite and complementary. In other words, the two attributes are not clear-cut, and restricting the meaning to only one side would obscure the other and detract from the diversity and flexibility of the Chinese character itself.

Secondly, many claim that the meditative interpretation of *zuowang* is appropriated from Zen Buddhism or later Daoist religion to interpret the *Zhuangzi*. Arguably the era of Zhuangzi had no direct relevance to later Zen or Daoist religion, but it cannot be deduced from this that the *Zhuangzi* itself does not contain quasi-meditation elements. From the development of religion history, in the Spring and Autumn and Warring States period, Zhuangzi's thought could have integrated meditation practices into itself, a possibility supported by the considerable literature and archaeological materials. For instance, Xiong Tieji 熊鐵基, in his research on "the Heaven-Worshipping Altar of *Shi Ji*" (*fengshanshu* 封禪書 of *Shiji* 史記), indicates that, after the Shang era of shamanism, there was an era of religion, which he argues in terms of the faith of the common people:

> "In the Warring States period, there were indeed a group of people who believed in and pursued 'the unity with the Dao' (*dedao* 得道) and to 'becoming an immortal' (*chengxian* 成仙) and developed various techniques of cultivation to attain the Dao. It can be said that the Daoist religion was born in the Warring States period." (Xiong 2021, p. 73)

Xiong also reminds us that the methods of internal alchemy have already been seen in the *Zhuangzi*, such as *xinzhai*, *zuowang*, and "exhaling the old and inhaling the new" (*tugu naxin* 吐故納新). The other evidence is represented in Roth's research on the excavated documents, and he identifies that many medical texts unearthed in Mawangdui 馬王堆 concern healthcare methods. In addition, he considers that, in the Warring States period, human healthcare methods, such as breathing and sitting contemplation, were widespread among different groups in society (Roth 1999, p. 179). It is clearly reflected in the early Daoist literature such as the *Laozi* and the *Zhuangzi*, and the meditative tradition of health preservation techniques constituted the historical origin of Daoism in the pre-Qin period.

The conventional interpretation tends to separate the experiential dimension of religious rituals from Lao-Zhuang philosophical Daoism, particularly evident in their rejection of the interpretation of sitting meditation. In contrast, Western sinology, especially Roth's decades-long engagement with textual archaeology, has made great efforts to highlight the core role of sitting meditation practices of the entire Daoist tradition, which has lasted for 2500 years and is still alive. One of his main aims is to restore the religious orientation of so-called "Daoist philosophy". The tendency he tries to correct is that the popular explanatory paradigm of Lao-Zhuang philosophy is heavily attached to the Wei-Jin Metaphysical style, especially the tradition of speculative commentary represented by Guo Xiang 郭象 (d. 312) (Roth 1998, p. 89). In Roth's view, Guo obscured the practical and technical dimensions of early Daoism and deliberately ignored the religious rites that appear in the *Zhuangzi*. Roth argues that this research paradigm splits the two dimensions of metaphysical and physical, which is not conducive to comprehend the original meaning of Zhuangzi's thought, as noted by those who concur with this viewpoint, such as Komjathy:

> "Although it is clear that there are 'philosophical dimensions' of Daoism, these are almost always rooted in a religious worldview, as well as in religious experience. In addition to philosophy, a nuanced understanding of Daoism must address cosmology, soteriology, theology, and so forth." (Komjathy 2013, p. 5)

In Komjathy's view, dividing early Daoism, which originally contained a few different dimensions, into the one-way interpretation of "philosophy" or "religion" is not in line with historical facts of the Daoism, and the dichotomous research paradigm of "demeaning religion and promoting philosophy" is not acceptable.

Lastly, from the central theme, we can find that the practice of sitting meditation is not against the idea of *wuwei* of Zhuangzi, but is one of the most important ways of "embodying the Dao". In Zhuangzi's case, the significance of meditation lies mainly in the methodological aspect, and practitioners could put this method aside when they reach the deep state of cultivation. As Schwartz has already shown, the ultimate goal of Zhuangzi's meditation is to reach the spiritual state of "unity with the Dao". However, the current view does not favor ritualistic or technique interpretations of *zuowang*. The main concern might be that it would lead to a misinterpretation of Lao-Zhuang philosophy, which is originally nonutilitarian, as a ritualism obsessed with secular life, especially for personal health and longevity. The typical examples are reflected by the criticism in Chapter 15 of "Constrained in Will" (*keyi* 刻意), such as "exhalation and inhalation" (*tuna* 吐納) and gymnastics of "guide and pull" (*daoyin* 導引). It seems that what Zhuangzi disapproves of is the obsession with physical immortality and excessive worldly pleasures, not a complete rejection of breath meditation as a method of nourishing the body.

It is my opinion that Zhuangzi can consider sitting meditation as a value-neutral technique to experience the Dao, and he neither advocates its transcendence nor denigrates its practicality. Sitting meditation should have been an everyday self-cultivation activity among Chinese ancients, even easier to start than the particular technique of "carving an ox". There are different levels of progressions of meditative cultivation in the *Zhuangzi*.[11] In the deep state of tranquility and emptiness, just as experiencing "merging with the Great Pervader", the practitioner is no longer bound to any particular techniques; Zhuangzi might liken it to "the fisherman forgetting the stakes" (*de yu wang quan* 得魚忘筌). Compared with the Confucian education of ritual and musical forms, the Daoist meditative way allows one to explore inward self-potential without resorting to language and logic, highlighting the importance of self-education. In short, people can achieve the goal of "nonaction" through the meditative method. This would be eventually manifested as an ideal way of cognition and interaction with the world, as outlined by the metaphor "take no action yet nothing is left undone" (*wuwei er wubuwei* 無為而無不為).

On the basis of the above analysis, one could obtain a more fundamental understanding of the religiousness of Zhuangzi's meditation. It is necessary to reveal the spiritual appeal behind it through the empirical phenomenon, and realize that Zhuangzi's ultimate spiritual pursuit not only lies in the earnest practice of the meditation ritual, but also goes

beyond the external regulation of the discipline to a certain extent, which directly taps into our hearts and minds. Moreover, the practitioner does not forget to return to this embodied psychophysical practice to gain the increasing power of efficacy. In doing so, one maintains a dynamic balance between the Dao (i.e., the fundamental) and the "techniques" (i.e., the incidental). In other words, Zhuangzi's discussion about the practice of self-cultivation, such as sitting meditation, belongs to the ontological level. This meditative experience is not only a particular cognitive schema, but also a unique human mode of being. It may be the utmost prominent feature of Zhuangzi-like meditation compared with primitive shamanism and later Daoist rituals.

The above explanation for accepting the interpretation of sitting meditation can combine the two interpretative paradigms of "sitting and forgetting" and "unintentional forgetting" into one, beyond the either/or approach. I think there is no clear-cut distinction between whether *zuowang* refers to an embodied practice technique or a unique mental or spiritual state. Zhu Tao (朱韜)'s doctoral dissertation combines two views of practical technique and mental state, opposing the dichotomous interpretation:

> "The practitioner must first practice 'sitting' before he can reach the state of mind of 'forgetting'. That is, 'sitting' is the premise of the optimal state of 'forgetting', and 'forgetting' is the final result of the cultivation method of 'sitting'." (Zhu 2021, p. 65)

The characteristic of "religion-is-philosophy" has existed since the pre-Qin period. Different schools of thought are all doctrines of self-cultivation to a certain extent, not purely metaphysical thinking, just as Roth points out. Lao-Zhuang's mystical writings come from a deep realization of their personal, practical, and embodied experiences. In contrast, the explanatory paradigm that simply devalues meditation practices and advocates the nonembodied spiritual cultivation may not be in accordance with the historical facts of the early Daoist school. This kind of spiritual or ethereal philosophy unrealistically exalts the cognitive level of human beings. It may make people indifferent to practical methods of body cultivation, and makes it difficult to experience the intersubjective knowledge and unity of theory and practice.

## 4. Clarifying the Uniqueness of *Zuowang* in View of Qi Transformation 氣化

The "indivisibility of the Dao and techniques" shows the unique feature of Zhuangzi meditative cultivation. Compared with the rites of the later Daoist religion, Zhuangzi's meditative forms are more diverse and flexible, and his language is more ethereal. In my opinion, the key to problematize the characteristic of Zhuangzi's cultivation lies in the Daoist's extraordinary insight into *qi* 氣. From its original meaning, the reference in the text to *qi*, which is clearly associated with meditation, generally pertains to the breath that circulates in human body. This kind of physiological *qi* basically conforms to the interpretation by inner alchemists. Nevertheless, Zhuangzi had a deeper understanding of this internal breath. He is keenly aware that the original meaning contains of the meaning of extension, which is succinctly summarized in Chapter 22 of "Knowledge Wandered North" (*zhi bei you* 知北遊) as "the one breath that is the world" (*tong tianxia yi qi* 通天下一氣) (Watson 1968, p. 235).

In Zhuangzi's unique vision, *qi* of meditative cultivation is not only the physiological breath that concerns the primary living conditions of humans and all lives, but also an objective vitality or vital energy similar to "pneuma", the ancient Greek word for breath or spirit. Regarding meditation practices, the physiological breath involved in breathing can be viewed as a special form of *qi* which dwells or inheres in the human body; the latter *qi* is regarded as the primordial or ancestral prior *qi*. The relationship between the latter and the former is similar to the Chinese philosophical categories of *ti* 體 and *yong* 用, i.e., the ontological existence of a thing and its application. Although cultivated breathing is not the prior *qi* itself, one can gain direct experience of *qi* in the primordial and ancestral sense by practicing breath meditation or other meditative cultivation. The prior *qi* can

be seen as the total root and the general basis of all things in the universe and, thus, can achieve the function of "permeating everything".

The prior *qi* manifests its dynamic, generative, and transformative characteristics through the two elemental concepts of the Yin 陰 and Yang 陽, as shown in Chapter 25 of "Sunnyside" (*zeyang* 則陽): "the Yin and Yang are breaths which are large 陰陽者, 氣之大也" (Watson 1968, p. 291). It is in accordance with this pair of basic concepts or modes that *qi* is revealed to have opposite and complementary forces or qualities. The dialectical relationship between the two, which are integrated into each other, promotes the movement, transformation, and development of *qi* and its self-harmony. The Yin and Yang modes of *qi* reflected in breath cultivation as the optimal living in which the "mind–body"(*shen-xin* 身心) maintains a dynamically balanced state. Just as Zhuangzi says, from "suffering from the Yin and Yang 陰陽之患" (in Chapter 4 of "The World of Men" 人間世) (Watson 1968, p. 58) to "the harmony of Yin and Yang 陰陽之和" (in Chapter 14 of "The Truning of Heaven" 天運) (Watson 1968, p. 156). In Chapter 13 of "The Way of Heaven" (*tiandao* 天道), the phrase "he and the Yin share a single virtue; in motion, he and the Yang share a single flow 靜而與陰同德, 動而與陽同波" (Watson 1968, p. 143) can be interpreted to mean that, by cultivating the mental state of the Yin and Yang within the mind–body, one can experience the harmony between the Yin side and Yang side of the prior *qi* prevalent in the universe.

Through the practice of breath-control meditation, the usual emotions and desires can be removed. The body serves as a natural tube, and the connection between the external world and the inner self can be restored. As the *Zhuangzi* goes, "the perfect man uses his mind like a mirror 至人用心若鏡" (in Chapter 7) (Watson 1968, p. 96) means letting the natural form of a thing itself be reflected in our mind effortlessly. Moreover, *qi* cultivation can bridge the gap between self-awareness and the physical body. The phrase "the true man breathes with his heels 真人之息以踵" (in Chapter 6) (Watson 1968, p. 77) indicates that every act of fully meditative breathing is the result of the coordination of the whole body, from top to bottom, from the body to mind. Gradually developing deep breathing habits will eventually lead to the realization that there is no crucial distinction between mind and body. Moreover, it leads to the experience that our body, which is not separate from the mind, is the infinite universe closest to the people themselves, such that every part of the body resonates with the movement of the universe as it is.[12] Every moment of exhalation and inhalation of the present moment brings one back to that infinite source of life, which enables one to be both in the world and beyond it.

Obviously, the interpretation of *qi* in Zhuangzi's cultivation should not only be limited to the empirical level of breathing but also contain insights into its ontological meaning. When one's physical body dies and stops breathing, *qi* does not disappear, but can continue to circulate objectively in the universe; as "The Great and Venerable Teacher" says, "that which kills life does not die 殺生者不死" (Watson 1968, p. 81). Specifically, I argue below that considering *zuowang* from the viewpoint of *qi* better elucidates the objective reality of mystical experience generated by meditation practices, as well as the commonalities of meditation with other forms of self-cultivation. It also highlights the specificity of Daoist techniques compared with other religious traditions, e.g., Christian meditation.

By practicing meditative techniques, one can achieve the mental state of "forgetfulness", and the occurrence of this kind of individual behavior has its objective necessity. In other words, the realization of "the Great Pervader" through meditation is not only necessarily substantiated in personal psychological experience but must also be logically affirmed. Fundamentally, the prior *qi* in the *Zhuangzi*, which permeates everything and keeps perpetually growing and changing, is the ultimate root for explaining meditative techniques and the objective basis for the unity between humans and the Dao. This vision of "qi transformation" (*qihua* 氣化) foregoes the subject–object dichotomy and instead shows that the private mystical experience does not deny its objective reality just because it is difficult to express and be experienced by outsiders. On the contrary, if the terms of meditation rituals are understood only from the viewpoint of personal psychological

experience, then the objectivity and intentionality of this ritual are seriously overlooked. Meanwhile, it is impossible to truly comprehend qi transformation as the internal completeness principle within Zhuangzi's theory of cultivation.

In my view, the interpretation of *zuowang* by Chinese scholars largely remains at the esthetic level of "unity with the Dao" attained through day-to-day activities, which is indeed influenced by a kind of interpretation based on the current Neo-Daoism. In contrast to this theory of spiritual state, there is the other extreme of the empiricism, which is also the point that many Chinese scholars might have misinterpreted the view of some sinologists. The empirical or reductive interpretation of Zhuangzi's meditation technique is obviously problematic because it only sees compliance with the phenomenon of meditation, without seeing the underlying foundation that provides its metaphysical basis. That kind of empirical interpretation deviates from the idea of "spontaneity" (*ziran* 自然) in the *Zhuangzi*. It fails to capture the characteristics of the indivisibility between "what is above form" (*xingershang* 形而上) and "what is under form" (*xingerxia* 形而下), and that "practice is reality" (*gongfu ji benti* 工夫即本體).

It is also important to clarify that meditation, like other nonmeditation techniques (e.g., carving an ox) in the *Zhuangzi*, can reach the profound state of "unity with the Dao". These are the cultivation practices that one does through the qi-transformed body in order to connect oneself with the outer world and the universe. However, the mainstream opinions on this issue rarely grasp the deep commonality between these two kinds of techniques in a comprehensive way. Rather, current studies tend to distinguish the differences between the two. For example, there is the distinction between the "internal and external" techniques pioneered by Zhong Zhenyu 鐘振宇, who regards "Huzi (壺子)'s demonstration of *qi*" and "Cook Ding carving an ox" as representatives of the two categories, without explicitly explaining that the two are the same in the ontological sense of qi transformation (Zhong 2014, p. 21). However, it seems to me that this distinction, if it must be made, is only on the superficial level. Both internal and external ways can lead to the mystical experience of unity with the Dao, and there is no need to distinguish between them.

Many scholars, such as Peng Guoxiang 彭國翔, emphasize that the difference between meditation practices in the Chinese cultural tradition (e.g., Confucianism) and other everyday activities lies in the expediency of the former and the fundamentality of the latter (Peng 2008, p. 24). My point of view is that they both can be the ultimate way to embody the Dao, and the distinction between "internal and external" is only for the convenience of academic expression or heuristic purpose. Particularly in the "spontaneity" view of Daoism, the various forms of "embodying the Dao" should not artificially impose a value hierarchy. As such, it can be understood that the *Zhuangzi* fully respects the variety of mystical experience forms in each state of existence. It complies with the specific historical situations in which the practitioners adapt themselves naturally. Maybe because of this, the *Zhuangzi* presents various modes of meditative cultivation. On the contrary, if we insist on making a hard distinction between the two types, i.e., breath meditation and other nonmeditation ways, it is against the spirit of "making all things equal" (*qiwu* 齊物) and would depart from Zhuangzi's idea of "facing natural guidelines calmly" (*anming* 安命).

Nevertheless, breath-sitting meditation shows its unique significance compared with other methods such as "carving an ox". This form of cultivation is decisive in the diachronic transmission of the early Daoist school, as fully demonstrated in Roth's study. Furthermore, meditation is more widespread than other forms of cultivation in the impact of the physical and mental nourishment. Indeed, the insights gained through the breath-meditation practices can sharpen one's intellect and, therefore, enhance one's mental acumen, and it is more effective in strengthening the body's health, eliminating diseases, and prolonging life. It focuses more on using our own body, which is the most personal and most accessible way for people to enter a state of deep tranquility and, therefore, most conducive to gaining enlightenment and spiritual enrichment. For ancient Chinese philosophers, the mechanism of the human body's function is based on the cosmic order of "Heaven and Man respond to each other" (*tian ren xiangying* 天人相應) or "Heaven and

Man are united as One" (*tian ren heyi* 天人合一). The mechanism of the meditative body closely relates to the overall balance of the energy of the Yin and Yang, as well as the harmony of the "Five Phases" (*wuxing* 五行) that pervades all beings. More than any other embodied techniques, the theory of breath meditation makes clear that the human body resonates with the whole universe.

Lastly, it can be argued that Zhuangzi-like meditation is qualitatively different from other religious cultures that involve meditation, mainly Western revelational religions. This difference is rooted in the different metaphysical foundations of each tradition, with "monism of *qi*"[13] as the basis of the Chinese tradition, while the Western tradition rests primarily on the metaphysics of substance with distinct entities. However, some scholars unconsciously interpret the meditation phenomenon from the viewpoint of Western-style dichotomous thinking, not realizing the characteristic of Zhuangzi-like meditation, which is probably one of the main reasons why they reject interpreting *zuowang* as an unearthly meditation technique of "sitting and forgetting" necessary for philosophical sense. Romain Graziani criticizes this misplacement, arguing that Zhuangzi's meditative cultivation differs from Western contemplation (Gong 2008, p. 88). In his view, the prevailing interpretation places the meditative body in the context of mind–body dualism. If applied directly to Zhuangzi's theory of cultivation, it would fall into a misinterpretation of Chinese thoughts from the lens of Western tradition. However, we already know that the body cultivated through Zhuangzi's breathing meditation is not the flesh body dominated by the mind in the dualism of mind and body, but the qi-transformed body, and the cultivated mind is no longer a rational mind distinguished from external objects, but the fasting or empty mind. In short, the meditative cultivation in Zhuangzi's vision does not seek to rid the soul of the physical body nor encourage self-renunciation of the human world in which people actively interact with others.

In general, Zhuangzi's theory of cultivation is based neither on Western thoughts nor on purely empirical explanations, but on the unique principle of qi transformation. He is particularly concerned with the role of the mind–body. From the perspective of qi transformation, we can gain insights into the specificity of Zhuangzi-like meditation, which differs from later Daoist meditation, as well as from Christian contemplation. Nevertheless, there is also a dimension of ultimate universal concern found in its particularity. As demonstrated below, a deeper meaning of Zhuangzi's religiousness can be enhanced if it is placed in the comparative religious perspective on different mystical traditions.

## 5. The Ultimate Concern of "the Unity of Self" in the Transformed Cultivation

The ultimate spiritual experience of "oneness" or "unity" with the Supreme Being or ultimate reality is achieved through meditation practices. In this regard, the religiousness of early Daoist cultivation can be seen to have some commonalities with other mystical traditions that focus on meditation methods, such as Christianity, Hinduism, and Zen Buddhism. Yang Rur-bin holds a similar view of the explanation of *zuowang*:

> "The special experience that Zhuangzi calls 'embodying the Dao' actually has significant elements shared with other cultures. Mystics of almost all cultural traditions and nationalities affirm that there is a higher self beyond the experiential self, and that one can be united with the higher reality. In the state of unity, the practitioners are complete and clearer, even though they have lost their everyday consciousness." (Yang 2016, p. 465)

Based on the premise of this commonality, the specificity of Zhuangzi's realization of the issue of "unity with the Dao" can be revealed compared to Western philosophy and religion.

The concept of *wang* within *zuowang* has an essential meaning in Zhuangzi's thought, which is the pivotal realization of "merging with the Great Pervader", i.e., "unity with the Dao". What does the mystical experience of "unity" or "oneness" by means of forgetting look like? Why does this form of Oneness have the ultimate significance for an individual development and process of transformation? The two "what" and "why"

questions are the keys to elucidating the religious implications of Zhuangzi's theory of cultivation and its ultimate concern.

For the mystical experience of early Daoism, humans and the Dao are a participatory, generative, and processual "becoming" rather than self-contained entities as in the Greek philosophical tradition. The formation of the "oneness" relationship between the two is not the external "union" of the Hebrew religion that commits man to God.[14] Humans and the Dao in Zhuangzi's mystical philosophy have the highly dialectical relationship of integration; when things are what they are, the Dao is naturally reflected in it, and the more individuals become themselves, the more they become integrated into the Dao and able to see things in the universal perspective of the Dao. In that sense, the mystical perspective can be said to be "seeing things as equal". The realization of individuality is formed by cultivating selflessness, just as the *Zhuangzi* says, "I lost myself" (*wu sang wo* 吾喪我). In other words, there is no self in the direct experience of the Dao, yet there is the authentic self (i.e., the self of selflessness) embodied in it. In Zhuangzi's mystical experience, there is no self, and there is the self, and the Dao is revealed in the self without the self.[15] This profoundly reflects Zhuangzi's dialectical relationship of "unity".

The Dao is the primordial root and basis of humanity and all beings. There is nonduality between the Dao and humans, and the Dao is just waiting for people to realize their natures or natural propensities through the way of "return" (*fan* 返) or "revert" (*fan* 反). Zhuangzi's thought returns us to the perspective of the Dao to inspire humans to see things in a holistic way. From this perspective, it expresses an unconditional respect for all beings (especially humanity) and insists that all are born with their own divine guideline of self-enlightenment and self-realization. It recognizes that this sacred guideline is flexible and not forcing people or things to be uniform. In the view of qi transformation, the nature of humans and the nature of things are not only equal but even consistent. "The Genuine Being" (*zhen ren* 真人) in Zhuangzi's writings, with a nonutilitarian esthetic attitude and childlike innocence, provides insights into the innate beauty within all life and gives voice to the realization that both humans and all living things constitute the entirety of the "qi-transformed" beings.

Nonetheless, there is a peculiar tendency in the *Zhuangzi* to go beyond the realization of the original nature of individual toward the self-realization of humanity. This self-reflexive stage of development is inevitable and essential. That is to say, the reflexivity of human nature is also the reflexivity of the universe itself and that self-realization is the realization of the universe. The uniqueness of human nature lies in the consciousness of "returning" or "restoration", from the trend toward self-deviation. This process of return involves the recovering of the original and natural self that has been altered, and restoring the wholeness and authenticity of our human potential. In short, it is realizing the ultimate ideal of "the unity of self". Zhuangzi's theory of cultivation shows that the self-transformation brought about by the mystical experience of "merging with the Great Pervader" is oriented by this "returning", and its transformational goal lies in "the unity of self (as the universal self)". On the basis of this transformative principle of "return", Lao-Zhuang Daoism reveals from the ontological perspective that there is a divine power in all beings, which the *Laozi* calls the "spirit-like vessel" (*shenqi* 神器) or what the *Zhuangzi* calls "the genuine being". They encourage people to reclaim that authenticity of nature because that is what we were given in the first place just as Durkheim emphasized. For Zhuangzi, a human is not like a solidified particle that only submits to the established reality and its given fate but is a being with an active, enterprising spirit. For human life, the moment of being thrown into the world with all that is given occurs at the same time as our preparation of returning and reverting. While various social, cultural, and physical forces condition the birth and growth of humans, there is also a metaphysical instinct and drive for "returning", which is the call of the Dao to people. The realization of the principle of "returning" by each person is also the fulfillment of the authentic self, i.e., the realization of "the unity of self".

Many scholars have realized that the mind–body practices of Zhuangzi's cultivation has profound effects of "transformation", and the foundation of this transformative mechanism lies with *qi*.[16] This "transformation" or "metamorphosis" mechanism is significant in understanding the intersubjectivity of oneself and others. It is shown in meditation practices, i.e., by transforming one's cognition and behavior, one can transform the way of viewing and treating external things, allowing each thing to live naturally. In other words, when people change their minds, they also change the world. The Dao of Zhuangzi is revealed in this dialectical unity between humans and the world. When I and all living things gradually recover our innate authenticity and wholeness, the inner self will eventually return to a deep connection with the source of the world, and to be the universal self, and everything has personal significance.

Zhuangzi's meditative cultivation aims to restore the naturalness of things by restoring authenticity humanity itself, i.e., to reveal "the unity of self" for all beings. The nature of self-unity is not completed once and for all, but is renewing, everlasting, and full of possibilities. In other words, it is the self-unity that is never finished and will ever respond to prompts of life. When we encounter various kinds of bondages again and again, Zhuangzi inspires us to go beyond what you already own. This state of self-unity which is a kind of "no self" is not caused by an external "other" but through achieving one's authentic self in the transformative cultivation.[17] In Zhuangzi's theory of cultivation, the apophatic expressions such as "transformation", "forgetfulness", "nonaction", and "emptiness" are all intended to reveal the authenticity and wholeness of the self. At the same time, this transformation effort actually involves a compassionate understanding of the limitations and defects of human beings, which supports each one to fulfill their natural form as they are. In this sense, it can be said that the transformation of "the self-unity" profoundly embodies the meaning of the religiousness of Zhuangzi's theory of cultivation.

In terms of the ideal of "the unity of self" of individual life, the religiousness of early Daoist philosophy has a profound universal human significance, comparable to other mystical traditions in terms of our ultimate concern. It has reached the ultimate level of self-transcendence and self-improvement of humanity. From the perspective of cross-cultural comparison, this is a very important religious dimension shared by early Daoism and other Axial civilizations. It shows the characteristics of the Axial era of human self-awakening: the common pursuit of a better, transcendent ideal of humanity. Meanwhile, Zhuangzi's ultimate concern of "unity with the Dao" is distinctive in its comprehension of the dialectical unity relationship between the Dao and humans. Thus, it is different from other mystical types as it is formed by its unique cultural identity. However, the religiousness of Zhuangzi's theory of cultivation is not arrived at in the context of narrow historicism since its specificity also reflects the core experience of human beings, with universal significance.

## 6. Conclusions

*Zuowang*, as an important representation of the Zhuangzi's cultivation, has been interpreted within two paradigms, that of Chinese scholars and that of sinologists. The first offers the argument that *zuowang* is a mental state with "de-meditation", and the other supports "sitting and forgetting" with breath-meditation practices. The former criticizes the latter against the original meaning of the *Zhuangzi*, suggesting that it could not reach the profound and consummate interfusion of "embodying the Dao", while the latter points out that the current interpretation is ethereal, is disembodied, and does not conform to the historical facts of the Daoist school. These two seemingly competing paradigms are, in my view, two sides of one thing. Their combination allows for the full understanding of the two sides of *zuowang*, namely, its methodological significance and the ontological significance, *yong* and *ti*, which are actually indivisible and are one.

The failure to see both sides of *zuowang* stems from the interpretations of a few of sinologists who neglected the philosophical significance of the mystical experience generated by meditation rituals, instead letting Chinese scholars to pay too much attention to the technical and experiential aspect of mysticism. An example of this would be mistaking

Roth's interpretation of *qi* as just physiological breath.[18] If we take *qi* as a pivot with an insight into its complexity and integrity, we gain a sympathetic understanding and resonate with their view. On the other hand, the mainstream view of Chinese scholars tends to follow and lean heavily on the traditional interpretative paradigm.[19] Its adherence to the dichotomy of "Daoist philosophy" and "Daoist religion" reveals a lack of self-awareness and self-confidence in one's own cultural identity, whether philosophical or religious. Meanwhile, most scholars do not approach embodied practice in their research,[20] this kind of disembodied methodology makes Daoist mystical philosophy more mysterious.

When we talk about the religiousness of Zhuangzi's philosophy, one should avoid parochial prejudices, since it cannot be characterized by the typical academic divisions of "religion" or "philosophy". The use of "religiousness" here does not suggest that Zhuangzi's thought falls within a certain definition of "religion", but the aim is to analyze the inner meaning and significance of the religious dimension of Zhuangzi's thought, so as to truly reveal the spiritual depths of its mysticism. As far as I am concerned, the interpretation that merely emphasizes the fact that the *Zhuangzi* contains empirical descriptions of meditative praxis and techniques does not uncover the true nature of the religious or mystical dimension of Zhuangzi's cultivation practices. The other interpretative paradigm of "de-meditation", which always remains at the level of the mental state of "unity with the Dao," also fails to reveal the essence of Zhuangzi's mystical philosophy. As Parrinder points out, it does not make much sense to claim that the experience of unity is the only important feature of mysticism (Parrinder 1976, p. 192). What is really important is to illumine the meaning of the mystical experience obtained through the mystical praxis of the method of *zuowang*, as well as its significance for the purification of the mind–body and the activation of human potential. This is precisely what is reflected in the realization of "the unity of self" in Zhuangzi's thought. With a firm grasp of this, it is achievable to construct Zhuangzi-like mysticism on the basis of the existing philosophical interpretation.

The profound religiousness of *zuowang* in Zhuangzi's cultivation lies in the humanistic relationship of Zhuangzi's mystical experience of "unity with the Dao" to the development of human potential. At this point, we see that Durkheim's redefinition of religion and Zhuangzi's ultimate concern have profound intercommunity. In the exploration of human potential, one can find that the religiousness of Zhuangzi's thought is both national and universal. The self-realization and self-cultivation of humanity is not only of utmost significance in the interpretation of the mystical element of Zhuangzi's thought, but it is the common basis for the communication between Zhuangzi's mystical philosophy and the world's mystical traditions. As such, it is of utmost significance for understanding the most precious religiousness of human beings.

**Funding:** This research received no external funding.

**Conflicts of Interest:** The author declares no conflict of interest.

## Notes

[1]     References to the text are made according to *Zhuangzi Yinde*. See (Hung 1956, 17/6/92-93). It also refers to *Zhuangzi jishi* 莊子集釋. See (Guo 2013, p. 259). Graham was consulted for translation purposes. See (Graham 1981, p. 92). Roth slightly deflects from Graham in his translation of *tong yu datong* as "merging with the Great Pervader", which seems more illuminating. See (Roth 2000, p. 37).

[2]     For examples of those identifying *zuowang* or *xinzhai* as "meditation technique", see (Ching 1983, p. 232), (Coutinho 2013, p. 209), (Graham 1989, p. 189), (Komjathy 2013, p. 206), (Maspero 1978, pp. 307–8), (Schwartz 1985, p. 218), and many works by Harold D. Roth. McConochie's doctoral dissertation (2017) provides one of the most comprehensive histories on the dimensions of mystical interpretation (not just on the technical aspect of meditation) by English-speaking scholars focusing on *zuowang* and *xinzhai*. For a more inclusive list of references, see (McConochie 2017, pp. 303–23).

[3]     Many Chinese scholars have focused on the significance of the comparison between mysticism, Daoism, and Chinese philosophy. Some of the more representative commentators include Feng Youlan 馮友蘭 (1895–1990), Yü Ying-shih 余英時 (1930–2021), Zhang Xianglong 張祥龍 (1949–2022), Cheng Lesong 程樂松, Zheng Kai 鄭開, Chen Shaoyan 陳紹燕, Bao Zhaohui 包兆會, and many Taiwanese scholars such as Yang Rur-bin 楊儒賓, Lai His-san 賴錫三, Carlo Kwan 關永中, and their students Wang Shuli 汪淑麗,

Lin Xiude 林修德, Xie Junzan 謝君讚, etc. Their research shows that "mysticism" can add a new perspective to understanding Zhuangzi's thought and facilitate fruitful modern academic conversations.

4     Maspero shows that the mystical practice in Zhuangzi's texts generally goes through the following stages: first, the rejection of worldly life, then the purification of self-consciousness, and then into the vacuous minds to become Oneness with the Dao, and finally, to "unify" all things in the way of the Dao. For more details, see (Ma 2022a, pp. 119–20).

5     Wen Yiduo 聞一多 (1899–1946) holds a similar view in 1941 about the religious origin (which he calls "ancient Daoism") of Lao-Zhuang's thought. He also considers the Daoism of Zhang Daoling 張道陵 (34–156) in the Eastern Han dynasty to be a later development of the same system as the "ancient Daoism" or even a revival of it, although the two were different in organization and form (Wen 1993, pp. 449–50).

6     Roth's list comprehensively presents a large number of important concepts in the *Zhuangzi* that are embodied in the process of breath-meditation practices. See "A Summary of Inner Cultivation Ideas" in (Roth 2021, pp. 243–44).

7     For a detailed discussion of Roth's Daoist contemplative practice, see (Roth 2021, pp. 238–45).

8     There are quite a lot of studies on Daoist body cultivation of health and longevity presented by Livia Kohn. She provides an engaging and insightful introduction of Internal alchemy tradition, as well as analyzes Zhuangzi's *zuowang* in the historical context of Daoist meditation practices. See (Kohn 2009, pp. 1–22).

9     As Fukunaga Mitsuji and other scholars point out, the main idea and characteristics of Zhuangzi's thought lie in "the transformation of perspective" (or "perspectivism").

10    Wu argues that, in most cases in the *Zhuangzi*, phrases containing the word 坐 describe a common sitting posture (e.g., 跪坐, 正坐, 匡坐, 安坐, and 休坐) in terms of good behavior and do not refer to a specific sitting posture, such as the meditation sitting posture with the special mental state of "forgetting". He also assumes that while there are descriptions of the meditation sitting posture, the only two cases of 坐忘 and 坐馳 are not typical in the *Zhuangzi*. In addition, he holds that, in other pre-Qin texts (such as the *Book of Rites*, the *Guanzi*, and his misquoted the *Selections of Refined Literature* of the Southern Dynasties), the phrases of 坐-X do not refer to any sitting posture. From this, he goes to the other extreme, interpreting 坐-X to mean "nature" or "naturally" (Wu and Huang 2017, pp. 39–40). Basically, I think that the reasoning presented by Wu falls into a false dilemma fallacy. The meanings of 坐-X (whether in the *Zhuangzi* or other ancient texts) are actually compatible with both the sitting posture and non-sitting interpretations.

11    In the *Dazongshi*, it is not clear from the text whether the practice of Yan Hui's forgetting of "benevolence 仁", "rightness 義", "propriety 禮", and "music 樂" (in the chapter of "In the World of Men") also involves a particular way of the body sitting postures. However, if we take into account similar scenes, such as the disciples' surprise at the sitting posture of his master Tzu-ch'i, who said, "The man leaning on the armrest now is not the one who leaned on it before 今之隱几者, 非昔之隱几者也" (Watson 1968, p. 36). It seems that the forgetfulness of benevolence, righteousness, propriety, and music is related to the quiet-sitting method of meditation, but the "forgetfulness" of Yan Hui seems to be of different contents of consciousness.

12    Robin's view of female Daoism also highlights the important role of "the lived body" in cultivating one's embodied subjectivity and identity. She elucidates the inseparability of the mind and the body, oneself and others, achieved through the practice of body transformation (e.g., Zhuangzi's fasting) in the vision "the flowing *qi*". See (Wang 2009, pp. 286–91).

13    I use "monism of *qi*" (not my phrase) to grasp the major characteristic of Zhuangzi-like meditative or mystical philosophy. Some readers may contend that the concept of "monism", appropriated from Western philosophy, as in Spinoza's metaphysical philosophy, is out of place here. However, in fact, this kind of monism is simply one of the two fundamental types of comparative mysticism, the other being theism. In this sense, Western mystical philosophy (e.g., Plotinus's conception of the One) can also be viewed as a form of monistic mysticism, but it is essentially different from the "monism of *qi*" of Zhuangzi. In addition, many scholars have clearly identified Zhuangzi's thought as a monistic form of mysticism, see (Yü 2014, p. 169) and (Lai 2011, pp. 27–32).

14    A typical view is that of Carlo Kwan 關永中, who discusses the mode of experience of Zhuangzi's philosophy as a kind of "up-and-down" relationship and "human heart returning to its origin" activity (Kwan 2002, pp. 113–15).

15    In view of the term "self" in the Zhuangzi, Jochim contributes a careful and provocative comparative survey from a linguistic perspective. He challenges the conventional interpretive paradigms that impose a modern philosophical concept of "self" or a "postmodern or Buddhistic rejection" of "self" (p. 68) on the *Zhuangzi*. For detailed cultural nuances of "self" among scholarly tendencies, see (Jochim 1998, pp. 37–46). Moreover, Jochim concludes that, with regard to Zhuangzi's philosophy, "it does not involve believing in but abandoning the (false) self in order that one can discover a deeper and truer no-self 'self'" (p. 68), which precisely coincides with my understanding of "the self without the self".

16    The transformation mechanism is well illustrated in Lai His-san's concept of "metamorphosis". It shows Zhuangzi's metaphors about the "metamorphosis" of all things, such as "the one finger of Heaven and Earth", "the one horse of all things", "the transformation of the Kun into the Peng", and "the materialization of the butterfly", all reveal the flowing view of qi transformation (Lai 2012, pp. 31–36).

17    In contrast with contemporary French philosophy, while some Chinese scholars have maintained that the "transformation" dimension of Zhuangzi's thought is similar to Gilles Deleuze's concept of "middle", Michel Foucault's "and", or Jacques Derrida's

"neither…nor…". I think that such a view does not seem to give full credit to Zhuangzi's unique dialectical view of "the self in the no-self" and "changed on the outside but not on the inside" (*wai hua er nei buhua* 外化而内不化) (Watson 1968, p. 46).

18   I must apologize for my inaccurate and one-sided comments on Roth's Daoist *qi*. For details, see (Ma 2022b, pp. 247–51).

19   As McConochie points out, many Chinese commentators of the *Zhuangzi* have the problem of "appeal to authority" in methodology (McConochie 2017, p. 269).

20   The revelation of Zhang Zailin 張再林 on this problem is thought-provoking. He reminds us that "the biggest mistake in the current academic understanding of Chinese cultivation theory (*gongfu lun* 工夫論) is to regard *gongfu* simply as 'introspective' cultivation (like inner spiritual cultivation or inner moral self-discipline), while completely ignoring the 'body' practices in the sense of 'inner and outer integration'." (Zhang 2018, p. 36).

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
