# Peer review of "The Religiousness of Cultivation in the Zhuangzi: “The Unity of Self” of Zuowang 坐忘"

_religions, doi:10.3390/rel14050612_

Round 1

Reviewer 1 Report

The article is very interesting and well-written. However, I would highly recommend to the author that she or he defines what is meant by religious and mystic. In the last chapter of the article, the author clearly explains the meaning of zuowang and I cannot relate the arguments to any religious or mystical dimensions. In my opinion it is very difficult to understand Zhuangzi's meditative praxis in this way. Therefore, in order to avoid misunderstanding, I would suggest that the author clearly defines how she or he comprehends religiousness and mysticism in Zhuangzi's philosophy. In other words, how exactly is the method of zuowang related to these aspects. Apart from that, the article is very insightful and important for the scholarship interested in Daoist, in particular, Zhuangzi's philosophy.  

Reviewer 2 Report

This is a very interesting article that tries to recover the importance of the religious dimension of Daoist and Zhuangzian cultivation practices from overbearing philosophical interpretations. It is founded on a comprehensive grasp of both English- and Chinese language literature and a thorough understanding of the source material.

I recommend it for publication. That being said, I also have some comments and suggestions for the author.

The introduction of the first of Wu Genyou's three points (regarding zuo 坐) should be clarified as it constitutes an important part of the discussion but does not provide enough information to the reader why zuo can be read as "natural" or "naturally."

On page 11, the author worries about misinterpreting Zhuangzian meditation in terms of Western religious and philosophical paradigms yet uses the phrase "qi-monism" thereby describing Zhuangzian cosmology or metaphysics in Western philosophical terminology (i.e. dualism and monism belong to the same Western philosophical language game).

The discussion of the"true self" also has the same problem as above since the idea of a true self is often taken from Buddhism to interpret the Zhuangzi. The author should refer to Chris Joachim's chapter "Just Say No to No-Self" in Wandering at Ease in the Zhuangzi.

The author should also refer to the work of Robin Wang and Livia Kohn for detailed discussions on the relation between Daoist cultivation and body.

The English is, for the most part fine, but there is need for minor revisions in both grammar and idiomatic usage of certain phrases.

For example, "Shaman(ism)" should not be capitalized; 楊儒賓's transliterated name should be Yang Rur-bin as that is how he writes it himself; there is no need for quotation marks for Chinese terms left in pinyin and when something is to be in quotation marks, the Chinese characters should go inside rather than outside the marks; on page 9, yin is misspelled as "yi"; qi should not be capitalized (i.e. not Qi); scholars from mainland China should not be referred to as "native Chinese scholars," simply referring to them as "Chinese" scholars is fine.
